# Current and Future Approaches in Management of Chronic Spontaneous Urticaria Using Anti-IgE Antibodies

**DOI:** 10.3390/medicina58060816

**Published:** 2022-06-17

**Authors:** Olguța Anca Orzan, Liliana Gabriela Popa, Mara Mădălina Mihai, Anca Cojocaru, Călin Giurcăneanu, Alexandra Maria Dorobanțu

**Affiliations:** 1Dermatology Department, “Carol Davila” University of Medicine and Pharmacy, 011461 Bucharest, Romania; olguta.orzan@umfcd.ro (O.A.O.); liliana.popa@umfcd.ro (L.G.P.); calin.giurcaneanu@umfcd.ro (C.G.); 2Dermatology Department, “Elias” University Emergency Hospital, 011461 Bucharest, Romania; 3Dermatology Department, University of Medicine and Pharmacy of Craiova, 200349 Craiova, Romania; anka_cojocaru@yahoo.com

**Keywords:** anti-IgE antibodies, omalizumab, ligelizumab, chronic spontaneous urticaria

## Abstract

Chronic spontaneous urticaria (CSU) considerably alters patients’ quality of life, often for extended periods, due to pruriginous skin lesions, impaired sleep, unexpected development of angioedema, and failure of conventional treatments in properly controlling signs and symptoms. Recent research focused on the development of new therapeutic agents with higher efficacy. Although the production of specific immunoglobulin E (IgE) antibodies against certain allergens is not a characteristic of the disease, treatment with omalizumab, a monoclonal anti-IgE antibody, proved efficient and safe in patients with moderate to severe chronic spontaneous urticaria uncontrolled by H1-antihistamines. Ligelizumab, a high-affinity monoclonal anti-IgE antibody, may also efficiently relieve symptoms of unresponsive chronic urticaria to standard therapies. This comprehensive review aims to present recently acquired knowledge on managing chronic spontaneous urticaria with new anti-IgE antibodies. We conducted extensive research on the main databases (PubMed, Google Scholar, and Web of Science) with no restrictions on the years covered, using the search terms “anti-IgE antibodies”, “omalizumab”, “ligelizumab”, and “chronic spontaneous urticaria”. The inclusion criteria were English written articles, and the exclusion criteria were animal-related studies. ClinicalTrials.gov was also reviewed for recent relevant clinical trials related to CSU treatment. CSU is a challenging disease with a significant effect on patients’ quality of life. Current therapies often fail to control signs and symptoms, and additional treatment is needed. New biologic therapies against IgE antibodies and FcεRIα receptors are currently under investigation in advanced clinical trials. We reviewed recently published data on CSU management using these novel treatments. The development of new and improved treatments for CSU will lead to a more personalized therapeutical approach for patients and provide guidance for physicians in better understanding disease mechanisms. However, some agents are still in clinical trials, and more research is needed to establish the safety and efficacy of these treatments.

## 1. Introduction

Chronic idiopathic urticaria or chronic spontaneous urticaria (CSU) is a debilitating disease that significantly impacts the quality of life. It is characterized by the development of wheals (hives), associated or not with angioedema for a period longer than 6 weeks, due to known or unknown apparent cause [1,2]. Wheals (hives) are superficial pruritic skin lesions, characterized by central swellings of various sizes, surrounded by reflex erythema, that usually persists for less than 24 h [1]. Angioedema is defined as an edematous process in the deeper part of the dermis, subcutaneous or mucous tissue that can last for up to 3 days [1]. It may be perceived as painful rather than itchy [1]. Unfortunately, the disease generally follows a prolonged course. Identifying a causative factor and finding the most suitable therapeutic option often pose a great challenge for physicians. The patients’ quality of life is considerably altered due to persistent, severe itching, impaired sleep, and associated secondary psychological and social issues [1,2].

Urticaria is considered a disease driven mainly by mast cells [1]. Symptoms develop due to mast cell and basophil degranulation, followed by the release of various types of mediators: Preformed (histamine, serotonin, tryptase, proteoglycans, etc.), newly synthesized lipid mediators (prostaglandins, cysteinyl leukotrienes, etc.), cytokines and chemokines (IL-4, IL-5, IL-6, TNF-alpha, TNF-beta, etc.) [3]. Mast cells and basophils activation may be immunoglobulin E (IgE) or non-IgE mediated. In addition, studies for other infiltrating cells involved in the pathophysiology of CSU, such as lymphocytes and eosinophils, are emerging (Figure 1). A significant role in type I allergic reactions has the platelet-activating factor (PAF) produced and released by mast cells, eosinophils, basophils, endothelial cells, neutrophils, platelets, fibroblasts, and even the cardiac muscle [4]. Mast cells can produce and be activated by PAF. When the mast cells are located in the skin, exposure to PAF leads to degranulation of their granules via neuropeptides [5]. Therefore, PAF plays an essential role in patients with urticaria due to its inflammatory role and chemotactic action. Along with the vascular endothelial growth factor (VEGF), PAF increases the permeability of capillaries in the skin and intensifies the development of urticarial specific lesions, such as wheals and erythema. This effect is especially distinguishable in chronic spontaneous urticaria. Studies on volunteers with CSU revealed that PAF injected subcutaneously induces typical urticarial hives [3]. Studies on anaphylaxis have shown that PAF is an important mediator in the development of anaphylactic shock. High serum levels of the platelet-activating factor directly influence the severity of systemic reactions [4,5].

Based on studies by Babaie et al. and Grieco et al. IL-6 plays a significant role in the pathogenesis of chronic urticaria by promoting the trans-signaling capacity during the inflammatory response [6,7]. IL-6 is released by mast cells, basophils, eosinophils monocytes, activated T cells, and neutrophils, particularly in acute urticaria resistant to conventional antihistamine therapy. Therefore, the disease activity could be explained by this chronic inflammation. Moreover, IL-6 is a major factor for the expression of other pro-inflammatory cytokines, such as TNF-alpha and IL1-beta, as well as for the production of antibodies [6]. Notably, the release of IL-6 precedes the appearance of antibodies in the serum [6,7].

Production of IgE plays a central role in the pathogenesis of allergic diseases. IgE is present both in the peripheral circulation and on the surface of different cell types, bound to low-affinity receptors, CD23 and high-affinity receptors, FcεRIα [4]. The binding of a particular allergen to specific IgE antibodies attached to high-affinity IgE receptors present on the surface of mast cells and basophils determines cross-linking and cellular degranulation. The released mediators produce cutaneous and, occasionally, systemic allergic manifestations [1,2,8]. In addition, various autoimmune conditions, including systemic lupus erythematosus, dermatomyositis, polymyositis or rheumatoid arthritis, have been associated with chronic spontaneous urticaria [9]. One large population study conducted on 12,000 CSU patients in Israel determined that female subjects have a higher incidence of systemic lupus erythematosus, rheumatoid arthritis, celiac disease, type I diabetes, and Sjögren syndrome than female patients without CSU [10]. The same analysis was performed between male subjects, but the numbers did not reach statistical significance. Further serologic investigations on autoimmune disease markers revealed that patients with CSU had significantly higher levels of antinuclear antibodies (ANA), rheumatoid factor (RF), anti-thyroid peroxidase (anti-TPO) antibodies, anti-parietal cell antibodies, antithyroglobulin antibodies, and anti-transglutaminase IgA antibodies [10].

This comprehensive review aims to present recently acquired knowledge on managing chronic spontaneous urticaria with new anti-IgE antibodies.

## 2. Materials and Methods

This non-systematic review aims to clarify the known data regarding anti-IgE biological therapy used for treating chronic spontaneous urticaria. We conducted extensive research on the main databases (PubMed, Google Scholar, and Web of Science) with no restrictions on the years covered, using the search terms “anti-IgE antibodies”, “omalizumab”, “ligelizumab”, and “chronic spontaneous urticaria”. The inclusion criteria were English written articles, and the exclusion criteria were animal-related articles. A total of 38 studies were included for review. We identified 22 studies containing the terms “omalizumab” and “anti-IgE antibodies”, and 16 articles containing the term “ligelizumab” between the years 2014–2022. ClinicalTrials.gov was also reviewed for recent relevant clinical trials related to CSU treatment.

## 3. Results

Anti-IgE therapy;Management of chronic spontaneous urticaria with omalizumab and ligelizumab in clinical practice;Potential biomarkers for the effectiveness of chronic urticaria treatment.

### 3.1. Anti-IgE Therapy

Chronic idiopathic urticaria or chronic spontaneous urticaria is a very challenging and frustrating disease for both patients and physicians. Consequently, different biological therapies have been developed as conventional treatments often fail to control the signs and symptoms [11]. During the past decades, research focused on both IgE antibodies and FcεRIα receptors as achievable targets for the development of new biologic agents that aim to prevent or decrease mast cell and basophil activation [12]. The correct selection of patients and the choice for the optimal method of investigation are essential for the proper evaluation of the effects of a newly developed treatment in any challenging disease. The objectives of a specific treatment also need to be clearly defined. Therefore, crucial information regarding a particular drug, the target population, and the disease itself can be obtained by evaluating the safety and efficacy of the drug in clinical trials and real-life scenarios. The gained knowledge helps us in developing solid criteria that enable physicians to formulate strict recommendations for the use of that specific medication.

The first licensed biological treatment for chronic spontaneous urticaria is the monoclonal anti-IgE antibody, omalizumab. It was approved for the treatment of mild to severe and long-lasting allergic asthma in 2003. In 2014, it was approved for the treatment of spontaneous urticaria for patients 12 years and older at a dose of 150 mg every 4 weeks and 300 mg every 4 weeks. This treatment primarily aims to achieve the relief of symptoms, such as pruritus and pain, as well as complete disappearance of the clinical signs in the least possible time frame. In addition, the treatment aims to ensure a better quality of life for chronic urticaria patients [1].

In accordance with the current international EAACI/GA^2^LEN/EuroGuiDerm/APAAACI guidelines for the management of chronic spontaneous urticaria, a four-step treatment algorithm is recommended (Figure 2). Following an inadequate response to the first and second lines of treatment with H1-antihistamines at licensed dose or up to 4 times the licensed dose, respectively, patients may be offered omalizumab as the third line of treatment [1]. Immunosuppressants, such as cyclosporine are administered to patients that are nonresponsive to omalizumab due to a higher incidence of adverse effects. Studies comparing the efficacy of omalizumab to cyclosporine showed the superiority of the first [13]. Moreover, in contrast to cyclosporine or montelukast, omalizumab is a licensed treatment for chronic urticaria [11].

In a series of randomized clinical studies, omalizumab administered subcutaneously in a monthly dose of 300 mg exhibited outstanding safety and efficacy in the treatment of chronic urticaria. A considerable decrease in urticaria signs and symptoms was achieved after 12 weeks of treatment compared with the placebo group [14,15]. Moreover, Zhao Z-T et al. [16] concluded that treatment with omalizumab is associated with a significantly higher reduction of the weekly itch severity score from baseline to 12 weeks compared with the placebo. Furthermore, the occurrence of angioedema and associated life quality impairments are well controlled with omalizumab in chronic urticaria patients [17]. Evidence from the last decade of clinical experience shows that treatment with omalizumab is efficient in more than 70% of patients suffering from chronic spontaneous urticaria [14,18,19] and is also of great benefit for patients with chronic inducible urticaria [20,21], even though it is not yet approved for its treatment. Omalizumab represents a symptomatic treatment, and its administration is recommended until the disease undergoes spontaneous remission, which generally takes 3 to 5 years [1,2]. Nevertheless, while many patients require continuous treatment for years, some patients only require a few months of treatment [22]. Variation in the response pattern is associated with patient-related factors and different phases of the disease in the same patient. Unfortunately, recurrence of urticaria symptoms is frequent 3 to 5 weeks after cessation of symptomatic treatment with omalizumab [23,24].

Notably, chronic spontaneous urticaria is not characterized by the production of specific IgE antibodies against common allergens [25,26]. However, the results of recent studies suggest that an autoimmune response, type-I or type-II, may play a role in the development of chronic urticaria [25,26]. The detection in these patients of IgE autoantibodies against various antigens, such as tissue factor or interleukin (IL)-24, staphylococcal exotoxins, double-stranded DNA, thyroglobulin, and thyroperoxidase, indicates a possible type I autoallergy or autoimmunity [25,26]. The ground for the hypothesis regarding the implication of a type-II autoimmune response is represented by the reports showing that a positive autologous serum skin test (ASST) leads to wheal formation at the site of the intradermal administration in approximately 30–40% of chronic urticaria patients. This may be partially explained by the presence of IgG autoantibodies against IgE or the high-affinity receptor (FcεRI) [25,26].

Given that 25–30% of patients suffering from chronic spontaneous urticaria, and an even higher number of patients diagnosed with inducible urticaria subtypes do not achieve complete response under treatment with omalizumab [19,25], there is still a need for safer and more effective therapies.

Ligelizumab has been refined from its forerunner antibody TNX901 and acts as a high-affinity monoclonal antibody neutralizing IgE [27], with an approximately 50 times higher affinity for IgE compared with omalizumab [28]. Several trials have investigated its efficacy in severe atopic dermatitis and asthma, both as intravenous and subcutaneous formulations [27]. However, the initial efficacy results of clinical studies did not support its use in these allergic diseases. On the other hand, in a multicenter, randomized, controlled phase II study, Maurer M et al. [29] proved the superiority of ligelizumab over omalizumab in the treatment of chronic spontaneous urticaria with a higher rate of responders and longer-lasting effect (recurrence of symptoms in 10 weeks compared with 4 weeks). In addition, chronic spontaneous urticaria is the only indication of ligelizumab.

There are currently six ongoing clinical trials on the efficacy, safety, and action mechanism of ligelizumab in adults and adolescents with chronic spontaneous urticaria, alone or compared with omalizumab [30,31,32,33,34]. More than 2000 patients are expected to receive a 300 mg dose of ligelizumab every 4 weeks. In addition, a placebo-controlled group will be switched from omalizumab to ligelizumab from week 24 to week 52. Ligelizumab is a potentially effective treatment for patients with refractory CSU, nonresponsive to omalizumab therapy prior to the use of immunosuppressants.

Other two phase 1 clinical trials are currently investigating the safety, tolerability, pharmacokinetics, and pharmacodynamics of UB-221, an anti-IgE monoclonal antibody, in patients diagnosed with chronic spontaneous urticaria [35,36].

Metz M et al. reported another anti-IgE biologic, GI-301 (GI Innovation), a long-acting IgE trap-Fc fusion protein that may exhibit higher and more durable binding to IgE compared with omalizumab [28]. However, data from randomized clinical trials are currently lacking.

### 3.2. Management of Chronic Spontaneous Urticaria with Omalizumab and Ligelizumab in Clinical Practice

At present, few questions still remain unanswered. The discrepancies between clinical trial results and those seen in everyday practice are vital for practitioners. For example, in some instances, a better response to omalizumab has been observed in real-life clinical settings compared with pivotal randomized controlled trials [14,15,16].

Real-life practical evidence has demonstrated that patients with chronic spontaneous urticaria may exhibit a rapid response to omalizumab. A retrospective evaluation of patients who underwent treatment with omalizumab outside of clinical trials showed a quick control of the symptoms, with 57% of patients achieving complete clinical remission within a week [37]. In addition, our personal experience and reports in the medical literature are in accordance with the mentioned study results conducted by Metz et al. [38]. Furthermore, relapses of chronic urticaria in patients formerly treated with omalizumab can be successfully managed with the same biologic agent. A retrospective study showed that most of the relapses occurred within 2 to 8 weeks after discontinuation of omalizumab treatment [39]. Once again, all patients achieved complete remission of urticaria signs and symptoms within the initial 4 weeks after reinitiating omalizumab at a dose ranging between 150 and 600 mg per month, with no adverse effects [39].

In another retrospective study conducted in Spain on 110 patients with incompletely controlled chronic urticaria, the administration of omalizumab resulted in a complete or significant response in 82% of the patients. Moreover, 6% of patients were able to discontinue the other administered drugs for chronic urticaria and remained free of symptoms under monotherapy with omalizumab [40]. After a treatment period that varied between 1 and 18 months, approximately 37% of patients achieved complete clinical remission and ceased administration of omalizumab. Out of these patients, 48% required reinitiation of omalizumab due to relapse of the disease, which once again induced a complete response in 90% of cases within 1 week to 2 months [40].

In accordance with the current international EAACI/GA^2^LEN/EuroGuiDerm/APAAACI guidelines for the management of chronic spontaneous urticaria, the recommended initial dose is 300 mg every 4 weeks. Dosing is independent of total serum IgE [1]. Patients who do not achieve significant results and improvements in symptoms during treatment with omalizumab at the licensed dose can increase the dose up to 600 mg, reduce administration intervals to 2 weeks or both [1]. Patients need to be informed that increasing the dose of omalizumab is currently off-label.

Ligelizumab, at a dose of 240 mg every 4 weeks, safely and effectively controls not only chronic spontaneous urticaria, but also other chronic urticaria subtypes, such as delayed pressure urticaria [21], solar urticaria [41], heat urticaria [42], cold urticaria [43], cholinergic urticaria [44] or symptomatic dermographism [45]. Therefore, ligelizumab has increased potential to be the best treatment not only for chronic spontaneous urticaria, but also for difficult-to-treat cases of chronic inducible urticaria. Randomized controlled trials on the efficacy of ligelizumab in the management of chronic inducible urticaria subtypes (cholinergic urticaria, delayed pressure urticaria, and cold urticaria) is highly encouraged. Currently, there are no existing biological treatments approved for these urticaria subtypes, in which off-label omalizumab use has shown significant benefit to date. However, it appears to be less effective than when administered for chronic spontaneous urticaria [21]. Whether ligelizumab is also useful in other diseases in which off-label treatment with omalizumab has proven successful, such as allergic rhinitis [46], difficulties in initiating specific immunotherapy [47], nasal polyposis [48], mastocytosis [49] and also bullous pemphigoid [50], systemic lupus erythematosus [51] or Morbus Morbihan [52] require clarification in the future.

In a comparison study between ligelizumab and omalizumab performed by Soong W. et al. it was noticed that ligelizumab has the ability to inhibit IgE binding to FcεRIα more effectively compared with CD23, thus being more efficient than omalizumab [53]. Moreover, after stopping the treatment, the median time of well-controlled disease was 28.0 weeks [53]. These results demonstrate a more prolonged treatment effect of ligelizumab compared with omalizumab. Furthermore, previous non- or partial responders to omalizumab experienced a >40% increase in complete response rates after ligelizumab treatment at a dose of 240 mg/4 weeks after 12 weeks [54]. In addition, ligelizumab is more efficient in controlling prolonged angioedema symptoms than omalizumab [55].

Recent data suggest that ligelizumab’s superiority may be due to differences in their IgE binding sites [56]. Since omalizumab preferentially inhibits IgE binding to the low-affinity IgE receptor CD23, ligelizumab demonstrated superior inhibition of IgE by binding to the high-affinity receptor, FcεRIα [56].

In conclusion, preliminary clinical data and recent reports demonstrate more prolonged suppression of IgE with ligelizumab compared with omalizumab.

### 3.3. Potential Biomarkers for the Effectiveness of Chronic Urticaria Treatment

The prediction of variations in the disease activity in response to treatment can be very helpful in improving treatment algorithms. Therefore, identifying biomarkers that can be used to predict the course of the disease and the extent of variation in the activity of chronic urticaria upon administration of a particular treatment would represent a significant advancement in providing an optimal individualized therapeutic approach. This is applicable for all the drugs currently used to treat chronic urticaria, such as omalizumab, leukotriene receptor antagonists, cyclosporine, and H1-antihistamines.

Among patients treated with cyclosporine, a positive serum basophil histamine release assay suggests that these patients are more inclined to respond to cyclosporine treatment than patients with a negative test [52].

For the moment, data on which and how biomarkers may anticipate the result of the other treatments are lacking, but research on the subject is ongoing. An example of the interest in the field is a recent retrospective study that included 41 patients with chronic refractory urticaria and concluded that the absence of basophil CD203c upregulation activity in the patients’ serum was correlated with the clinical response to omalizumab [57]. The upregulation activity of CD203c was present in 18 out of 41 patients [57]. Interestingly, only 50% of these 18 patients recorded clinical improvement with omalizumab [57]. On the other hand, 87% of the 23 patients who did not display upregulation activity of CD203c had a positive clinical response to omalizumab treatment [57].

Grieco et al. discovered higher levels of IL-6 and IFN-gamma in patients with chronic spontaneous urticaria compared with those in the control group [6]. A decrease in disease activity and IL-6 and IFN-gamma levels was also observed following omalizumab treatment. Moreover, an increased level of IL-6 and IFN-gamma was noticed during CSU relapse phases. Levels of other tested cytokines were similar between patients with CSU and the control group [6]. Therefore, IL-6 and IFN-gamma have the potential to be valuable biomarkers for disease type and severity in chronic spontaneous urticaria.

In support with the perspective described above, biomarkers require further evaluation in prospective studies due to the strong potential to be valuable, clinically significant, response indicators to specific treatments.

## 4. Discussion

Several other potential therapies for chronic spontaneous urticaria are currently under investigation in phases 1, 2, 3 and 4 clinical trials.

Monoclonal antibodies against inflammatory cytokines released by mast cells, such as interleukin-5 (IL-5), the IL-5 receptor (IL-5R) or IL-4R have turned into recent research targets for CSU treatment [58]. In addition, the activity of the β-tryptase protease, also secreted by mast cells, can be inhibited by monoclonal antibodies, such as MTPS9579A, which is currently in phase 2 clinical trial for patients with CSU refractory to antihistamines [59]. Following the example of omalizumab, which disrupts mast cell activation by targeting immunoglobulin E, leading to IgE crosslinking and aggregation of FcεRI receptors, it was noticed that IgE crosslinking could also be inhibited after it has bound to FcεRI receptors using small-molecule Bruton’s tyrosine kinase (BTK) inhibitors. Bruton’s tyrosine kinase (BTK) is a non-receptor cytoplasmic kinase that plays an important role in oncogenic signaling in many B cell malignancies but is also involved in signaling through the Fc receptors and Toll-like receptors. Taking into consideration the extensive range of effects of Bruton’s tyrosine kinase, a focus has been placed on developing specific BTK inhibitors for the treatment of different diseases, from malignancies to atopic and chronic inflammatory disorders [60]. Therefore, small-molecule BTK inhibitors, such as fenebrutinib and remibrutinib are currently in stage 2 and 3 clinical trials for chronic spontaneous urticaria [61,62].

In recent years, new approaches in disrupting pro-inflammatory mechanisms have been explored. It was observed that the total number of mast cells could be depleted by targeting the stem cell growth factor receptor-tyrosine kinase KIT [63]. A new generation of small-molecule inhibitors against tyrosine kinase KIT receptors, such as CDX-0159, an anti-KIT monoclonal antibody, is currently in phase 1 clinical trial for chronic urticaria [63].

Due to the potential side effects and immunoreactivity [12], these promising future treatments must be approached cautiously, and continuous research is needed. Additional biologics and therapies currently tested are summarized in Table 1.

## 5. Conclusions

Chronic spontaneous urticaria has a significant medical impact worldwide due to its increasing prevalence in a general, age-dependent population, the potential to associate systemic symptoms and life-threatening angioedema, and due to the unpredictable evolution, particularly in patients with uncontrolled disease. Therefore, extending the therapeutical possibilities in monotherapy or integrative approaches is a priority. In the last decades, research focused on developing monoclonal anti-IgE antibodies. The first monoclonal anti-IgE antibody was omalizumab, which is now recommended by EAACI guidelines for patients with refractory urticaria. A second monoclonal anti-IgE antibody is ligelizumab, possibly with a higher IgE affinity than omalizumab. Additional recent agents are still waiting for evaluation by several ongoing trials. Although progress has been noted through the addition of novel therapeutic approaches, we believe that more insight and research are required on IgE antibody modulation correlated with the clinical response of CSU. Certainly, new biologic therapies and possible predictive biomarkers will be available in the future, allowing medical professionals to prescribe innovative and better treatments for patients struggling with chronic urticaria.

## Figures and Tables

**Figure 1 medicina-58-00816-f001:**
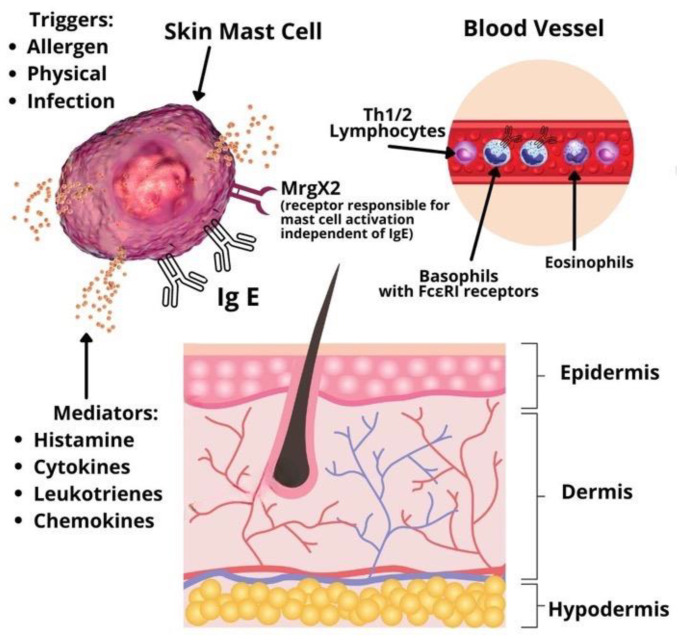
Pathophysiology of chronic spontaneous urticaria.

**Figure 2 medicina-58-00816-f002:**
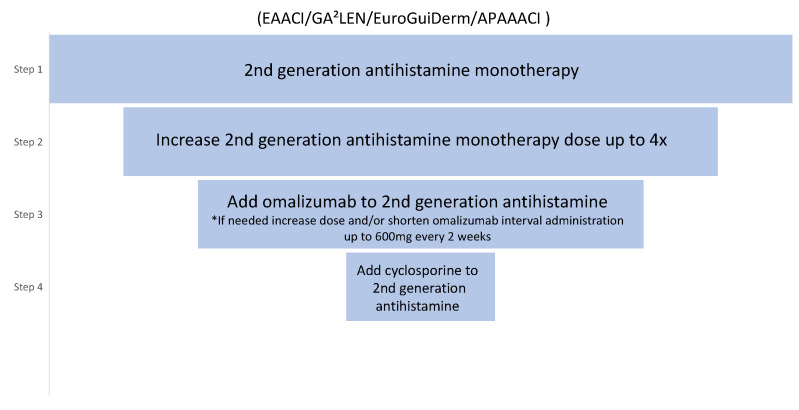
The international EAACI/GA^2^LEN/EuroGuiDerm/APAAACI guideline for CSU treatment. * Based on expert consensus and achieved >70% agreement in the consensus conference [1].

**Table 1 medicina-58-00816-t001:** Therapeutic agents currently under investigation in clinical trials for the treatment of chronic spontaneous urticaria, May 2022.

Molecule	Mechanism	Type	Stage	Indication
CDX-0159	Anti-tyrosine kinase KIT	mAb	1	CSU and chronic inducible urticaria [63]
Mepolizumab	Anti-IL-5	mAb	1	CSU [64]
Ligelizumab	Anti-IgE	mAb	2	CSU in children from 12 to <18 years of age [65]
Dupilumab	Anti-IL-4/-13	mAb	2	CSU [66]
MTPS9579A	Tetrameric β-tryptase	mAb	2	CSU [59]
Tezepelumab	Thymic stromal lymphopoietin	mAb	2	CSU [67]
Remibrutinib	Anti-BTK	Small molecule	3	CSU [68]
Benralizumab	Anti-IL-5 receptor	mAb	4	CSU [69]

Abbreviations: KIT: Tyrosine kinase; IgE: Immunoglobulin E; IL: Interleukin; BTK: Bruton tyrosine kinase; CSU: Chronic spontaneous urticaria; mAB: Monoclonal antibody.

## Data Availability

Not applicable.

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
