# Peer review of "Current and Future Approaches in Management of Chronic Spontaneous Urticaria Using Anti-IgE Antibodies"

_medicina, 2022, doi:10.3390/medicina58060816_

Round 1

Reviewer 1 Report

The manuscript ”NOVEL APPROACHES IN MANAGEMENT OF CHRONIC SPONTANEOUS URTICARIA USING  ANTI-IgE ANTIBODIES“ presents a review on this specific subject, based on the wide literature data and analysis of clinical study results.

However, I have some suggestions to improve this text of the review:

-Title: You write “novel” approaches, but you write data about current therapy with omalizumab.

- Please write “mast cells and basophiles” - instead of “basophiles and mast cells” (mast cells are more important for this disease)

- Please mention data from new article - by Zuberbier T, Abdul Latiff AH, Abuzakouk M, et al, 2022 - which presents the latest guidelines for CSU and please modify your data according them (The international EAACI/GA²LEN/EuroGuiDerm/APAAACI guideline for the definition, classification, diagnosis, and management of urticaria. Allergy. 2022 Mar;77(3):734-766. doi: 10.1111/all.15090.)

- Also, very useful is a reference by Johal and Saini (Johal KJ, Saini SS. Current and emerging treatments for chronic spontaneous urticaria. Ann Allergy Asthma Immunol. 2020 Oct;125(4):380-387.)

-Table 1 was based on data analyzed.. (when?  e.g. May 2022). Also, data mentioned in the table should be written according to the logical order (according to the number of the stage/ phase of clinical study; or according to alphabetic order;  or similarity in therapy, etc…

Abbreviations: please add it for KIT

-Figure 1.- Is it original or based on some article?

- Instead “Omalizumab, Ligelizumab” write a “omalizumab, ligelizumab”

-  Cited references are somewhat written in different manner – please check them to be written according to the same journal rules.

Author Response

Esteemed reviewer, thank you very much for your valuable suggestions. Here are the changes that have been made accordingly to your requests.

Reviewer 2 Report

Congratulations for the authors for their work

The authors are invited to ad in the introduction part between mediators Platelet activating factor -PAF and discuss.

Also the IL 6 action should be emphasised by the multifactorial expression of IL 6.Data on IL 6 release during COVID and its influence on eruptive urticarian flare ,if available may be added and referenced

Authors are invited to add data,if available on Multiple Autoimmune Syndorme and Urticaria and refference them.

Comparative data between the use of Omalizumab/Ligelizumab each one alone comparative to their association with 4 dose of antihistaminics/or ciclosporine are of interest for the clinical practice.

Author Response

Esteemed reviewer, thank you very much for your valuable suggestions. Please see the attachment.

Reviewer 3 Report

This study would benefit from more precise description of current Oma usage rules acc. to actual EAACI guidelines (2022)

Author Response

(The authors gave the same response as above.)

Round 2

Reviewer 1 Report

The suggestions are accepted.

Reviewer 3 Report

no more comments